# Repetitive Sequence Barcode Probe for Karyotype Analysis in *Tripidium arundinaceum*

**DOI:** 10.3390/ijms23126726

**Published:** 2022-06-16

**Authors:** Jin Chai, Ling Luo, Zehuai Yu, Jiawei Lei, Muqing Zhang, Zuhu Deng

**Affiliations:** 1National Engineering Research Center for Sugarcane, Fujian Agriculture and Forestry University, Fuzhou 350002, China; CJ1152648@163.com (J.C.); luoling2255@163.com (L.L.); ljw1124988684@163.com (J.L.); 2Key Lab of Sugarcane Biology and Genetic Breeding, Ministry of Agriculture, Fujian Agriculture and Forestry University, Fuzhou 350002, China; 3State Key Laboratory for Protection and Utilization of Subtropical Agro-Bioresources, Guangxi University, Nanning 530004, China; arhuay_yu@163.com (Z.Y.); zmuqing@163.com (M.Z.)

**Keywords:** FISH, karyotype, sugarcane, *Tripidium arundinaceum*, repetitive sequence

## Abstract

The barcode probe is a convenient and efficient tool for molecular cytogenetics. *Tripidium arundinaceum*, as a polyploid wild allied genus of *Saccharum*, is a useful genetic resource that confers biotic and abiotic stress resistance for sugarcane breeding. Unfortunately, the basic cytogenetic information is still unclear due to the complex genome. We constructed the Cot-20 library for screening moderately and highly repetitive sequences from *T. arundinaceum*, and the chromosomal distribution of these repetitive sequences was explored. We used the barcode of repetitive sequence probes to distinguish the ten chromosome types of *T. arundinaceum* by fluorescence in situ hybridization (FISH) with Ea-0907, Ea-0098, and 45S rDNA. Furthermore, the distinction among homology chromosomes based on repetitive sequences was constructed in *T. arundinaceum* by the repeated FISH using the barcode probes including Ea-0663, Ea-0267, EaCent, 5S rDNA, Ea-0265, Ea-0070, and 45S rDNA. We combined these probes to distinguish 37 different chromosome types, suggesting that the repetitive sequences may have different distributions on homologous chromosomes of *T. arundinaceum*. In summary, this method provide a basis for the development of similar applications for cytogenetic analysis in other species.

## 1. Introduction

Sugarcane (*Saccharum* spp.) is an annual or perennial C4 plant that is indigenous to tropical and subtropical regions and mainly used for sugar production and as a clean energy substrate [1]. The genus *Saccharum* and its related wild genus, including *Miscanthus*, *Sclerostachya* (Hack) *A. Camus*, *Erianthus Michaux*, and *Narenga porphyrocoma* (Hance) *Bor*, constitute the “*Saccharum* complex” [2]. These species are important wild germplasm resources to broaden the genetic base of sugarcane breeding [3]. *Tripidium arundinaceum* belongs to the genus *Tripidium* [4], which is one of the research hotspots for enhancing stress resistance in sugarcane breeding [5,6].

*T. arundinaceum* has the traits of drought tolerance, strong disease resistance, and wide adaptability. It is used as a parent material for basic hybridization by sugarcane breeders [7]. In the 1970s, the F_1_ generation of the cross between Badila and *T. arundinaceum* was produced in the Hainan sugarcane breeding station in China, but the second generation of the hybrid could not be bred due to the sterile F_1_ pollen. Until 2001, the bottleneck was broken by changing the methods of breeding [8]. Nowadays, a batch of excellent BC_3_, BC_4_, and BC_5_ materials has been bred [9]. Therefore, through the hybridization of sugarcane with the related genera, new varieties with high-yield and high-quality characteristics can be created, which are of great significance to sugarcane breeding [10]. Many researchers have successively used them as research materials to identify resistance, molecular identification, and chromosome genetics [11]. Meanwhile, the complex chromosomes, morphologically similar chromosomes, and poor chromosome markers make cytogenetic research difficult in *T. arundinaceum* [3,12].

Fluorescence in situ hybridization (FISH) is a cytogenetic technology in which the specific fluorescent-labeled probe is denatured and paired with the target sequence of chromosome through base complementarity pairing [13]. At present, FISH has been widely used in karyotype analysis, physical map construction, and genetic relationship analysis in plants [14]. With the development of cytogenetics technology, the application of large single-copy and low-copy probes has become more and more widespread [15]. Through GISH with the genomic DNA, D’Hont [5] and Georgy [6] found that the F_1_ is inherited in the way of *n* + *n*. However, the current research on chromosome inheritance focuses on quantitative changes, and there is no more precise chromosome identification technology for further precise research. Previous karyotyping studies based on the conventional technology and FISH with rDNA genes showed a high dependence on chromosome morphology and had a low degree of precision because of inadequate chromosome markers in *T. arundinaceum* [16]. These problems may impede our understanding of *T. arundinaceum* genome organization and evolution. Therefore, we look for repetitive sequences located on specific regions of chromosomes.

Contrary to the traditional in situ hybridization, Reverse Dot Blot (RDB) technology puts the probes dotting on a nylon membrane, makes these hybridized with the PCR products of the sample, and then observes the blot results of the hybridization through elution, antibody binding and color display [17]. This method can solve the disadvantage of conventional hybridization detection of a single sample and can detect the homology of multiple samples at the same time, which greatly improves the experimental efficiency [18]. Fritz [19] analyzed the variation in feeding host types of *Ant. gambiae* by quantifying responses by RDB. Huang Rongxian [20] used RDB to analyze the expression of *MF6*, which is a related gene that controls rapeseed fertility.

The content of medium and high-copy repetitive sequences is more than that of single copy sequences in plant genomes [21]. High-copy repeats renature is faster than low-copy sequences under the same condition. By interrupting the genome sequence, single-stranded DNA can be renatured into double-stranded DNA in plants. Therefore, the required repeats containing different copy numbers can be enriched, which are Cot-enriched repeats [22]. Cot enrichment of repetitive sequences is the fastest and most efficient for non-model plants without reference genomes [23]. Cot-enriched repetitive sequences are also used as probes in FISH, and most of the probes used in plant karyotyping research are mid to high-copy repetitive sequences [24]. In 2005, Wei used the Cot-1 DNA of *Brassica napus* as a probe to perform FISH and published chromosome karyotype with a marker for each chromosome for the first time [25]. Since then, Cot technology has been used to enrich repetitive sequences and mapped plant chromosomal karyotypes in different plants.

In this study, we construct a Cot-20 repetitive sequence library of *T. arundinaceum*. Screening the library by FISH, we can analyze the distribution of medium and high-repetitive sequences in *T. arundinaceum*. The study of repetitive sequences is of great significance for exploring the origin and evolution and revealing the infiltration of genetic material in the process of polyploidization.

## 2. Results

### 2.1. Cloning of Repetitive DNA Library from T. Arundinaceum

gDNA of *T. arundinaceum* was extracted using a CTAB technology, and a clear single band was detected for the digestion (Lane 1 and 2 in Figure 1). gDNA digested by DNaseI appeared 50~1500 bp dispersed painting, and it focused on 100~300 bp (Lanes 3 to 4 in Figure 1).

To obtain a suitable renaturation time, Cot-1, Cot-20, Cot-60, and Cot-100 were used for screening repetitive sequences. Reverse Dot Blot (RDB) was performed using *T. arundinaceum* gDNA as a probe and the 45S rDNA plasmid was a positive control. We can find different renaturation times for enriching different copy numbers of repeat sequences. The shorter the renaturation time, the more medium and high copy numbers are obtained. The results showed that Cot-1 enriched with the highest number of high copies, and Cot-20 enriched with the highest number of medium copies. The results of FISH showed that the high-copy sequence was located at both ends of all chromosomes and that intermediate-copy sequences showed a high degree of diversity, whereas the low-copy sequence showed no signal (Figure 2). Therefore, we selected the Cot-20 library for subsequent study based on the suitable renaturation and sequence of copies.

### 2.2. FISH and Blast for Barcode Selection

The Cot-20 library was enriched for a large number of the *T. arundinaceum* genome, and 1350 pre-probes were cloned for further FISH localization analysis. These probes were classified into nine types according to their location on the *T. arundinaceum* chromosome. Among them, 727 fragments showed signals at both ends of all chromosomes (Figure 3A); 163 fragments showed signals at one end of partly chromosomes (Figure 3B); 286 fragments showed no or low signals (Figure 3I). The other types were varied and located at telomeres, centromeres, or diffusely distributed (Figure 4). The results indicated that the Cot-20 library provided sufficient chromosomal markers for further chromosomal identification. Finally, the barcode probe consisted of six probes from FISH and blasted to sorghum, 45S rDNA, and 5S rDNA (Appendix A).

### 2.3. Karyotyping Analysis Based on the Barcode

Each barcode of the repetitive sequence will be labeled with either biotin (green) or digoxigenin (red), respectively. A group of two probes was hybridized in the same metaphase cell at each repeated round of hybridization. According to the combination regions and intensity of signals (Figure 5), we can find that they have weak signals, obvious signals, and diverse signal sites. Each type of chromosome was labeled by the specific barcode; then, we sorted the difference.

The probe Ea-0907 (green) was signaled on various types of regions, in which some signals were located at one end or both ends, and centromere or ends. Simultaneously, Ea-0098 and 45S rDNA (red) were located at the centromere-proximal regions (Figure 6). Through the statistics of the results (Table 1), we found that chromosomes can be divided into 10 types based on the characteristic barcode (Ea-0907, Ea-0098, 45S rDNA). However, we probably only marked eight sets of homologous chromosome karyotype (2, 3, 4, 5, 6, 7, 8, and 9 types). Type 5 has only five chromosomes with a strong green signal at one end and a weak green signal at the other end. Similarly, type 8 has only five chromosomes with both ends and a weak green signal in the middle. Furthermore, there are 12 chromosomes with a strong green signal at one end in type 1 and two chromosomes with a green signal at the middle in type 10.

### 2.4. Chromosomal Fingerprint of T. Arundinaceum Revealed by Multiple FISH

Further on, through three rounds of fluorescence in situ hybridization, seven probes of different repetitive sequences were positioned on the same metaphase cell, and the chromosomes were numbered (Figure 7). Among them, the positions of the upper and lower ends of the same chromosome were all uniform (e.g., the upper part on the left, the lower part on the right in the slanted chromosomes). The probes used for the first hybridization were Ea-0663 (red), 5S rDNA (green), and 45S rDNA (green). The probes for the second hybridization were Ea-0267 (green) and Eacent (Probes located at the centromere of *T. arundinaceum* chromosome are called Eacent) (red). The probes for the third hybridization were Ea-0070 (green) and Ea-0265 (red).

According to the statistical results (Figure 8A), the 60 chromosomes of *T. arundinaceum* were classified into 37 different marker types by seven probes. Among them, there are 23 marker types that can distinguish as a single chromosome separately; two chromosomes in eight groups shared one marker type (9 and 38; 13 and 17; 14 and 59; 15 and 60; 18 and 54; 24 and 50; 34 and 36; 55 and 58), three chromosomes in three groups shared the same marker type (12,23,43; 16, 22, 45; 28, 30, 46), and four chromosomes in three groups shared the same marker type (7,42,44,57; 21,33,35,53; 25,29,39,49).

5S rDNA and 45S rDNA were used to analyze to assess the ploidy level. We found that each special FISH signal barcode was detected on the homology chromosomes with a 5S rDNA signal in *T. arundinaceum* (Figure 8B). This random distribution indicated that the homology chromosomes also have distinctions. Meanwhile, the six types were detected on homology chromosomes with 45S rDNA signals in *T. arundinaceum* (Figure 8C).

## 3. Discussion and Conclusions

FISH is an indispensable technique in molecular cytogenetics and genetics [26]. Compared to other methods, FISH will need less cost and provide direct evidence involving cytogenetic and genome research. Compared to diploid species, polyploidy has a different way of genome evolution and a higher level of extensive gene expression. Polyploid plants have a stronger ability to adapt to the environment. The additional effects of polyploidy and expression mechanism provide value for crop improvement, evolution and inheritance. The polyploid genome abounds mass high-copy repetitive sequences, such as transposons, retrotransposons, telomere repeats, satellite DNA, etc. [21]. Polyploid events, caused by whole genome duplications (WGDs), are often accompanied by the tandem duplication of repetitive sequences, the repeated segmental duplication, or retrotransposition in plants. [27]. This explosive transformation could be caused by the variation of regulation or modification of epigenetics [27,28]. The rapid advance of the genome and molecular cytogenetic developed mass methods in designing and screening probes. Unfortunately, the contribution of repetitive sequences is uncertain. Based on the polyploid or allopolyploid genomes, the expensive price of probe synthesis and genome sequencing impedes the development of FISH, and more important is the disordered distribution of repetitive sequence [28,29]. For studying the function of repetitive sequences, researchers have developed a variety of methods, such as cDNA libraries, methylation filter libraries, and cot enrichment. The barcode probe of the repetitive sequence is a strong and effective method, which can reveal the regions of repetitive sequences, the copies of species, and the chromosomal karyotypes [30]. Meanwhile, the repetitive sequences enriched by Cot DNA can be used as blocking DNA during hybridization and as the repetitive sequence libraries of barcode probes yet.

It is of great significance for tracking chromosome inheritance and identifying the chromosomes in *T. arundinaceum* [31]. The non-homologous chromosomes inherited in the generations can be effectively distinguished, and the origin and composition of the chromosomes can be displayed [32]. In this study, by a screening of the Cot-20 library, the repetitive sequence barcode probes were used to distinguish eight chromosomes of *T. arundinaceum*. Due to the lack of the other barcode probes that can be labeled on the chromosome arm or that can mark a small number of signals individually, the karyotype of the whole *T. arundinaceum* genome has not been established. This method still provides a reference for the genome on molecular cytogenetic of the polyploid or allopolyploid plants.

Sugarcane is not only an important sugar-yielding crop, but it is also the representative of the polyploid of high copies model. Crossing with the wild species, resistance can be availably introgressed to sugarcane. *T. arundinaceum*, a *poaceae* of neo-polyploid species, as a breeding wild species with good resistance to sugarcane, undergoes the polyploidization event that evolved from diploid ancestors [27,33,34]. However, during this process, it is uncertain whether rearrangements or other parental chromosomes merge between homologous. Even in previous studies, the repetitive sequences under homologous chromosomes differed in diploid groups or sub-genomic groups, such as sorghum [35], potato [36], miscanthus [37], strawberry [38], etc. However, for neo-polyploid, it is an enigma still. We analyzed the different copy numbers from the Cot-20 library and then revealed the differences in repetitive sequence among homology. Using repeated FISH of the same chromosome, 37 special barcodes were located from seven repetitive sequence probes and the difference of repetitive sequences on the *T. arundinaceum* homology chromosomes. These differences may imply that under the polyploidization events, among homology, duplication is often accompanied by variation of the difference ratio to the expansion of gene or the random insertion of the transposon, etc. [39,40]. That as a possible reason leads to diploidization or a pre-subgenome.

## 4. Materials and Methods

### 4.1. Plant Material and DNA Extraction

*T. arundinaceum* (HN 92–105, 2*n* = 60) was provided by Hainan Sugarcane Breeding Station and grown in the greenhouse at Fujian Agriculture and Forestry University. Leaf tissues from these materials were ground in liquid nitrogen and stored at −80 °C. Total genomic DNA (gDNA) was extracted following an improved CTAB methodology [41].

### 4.2. Repetitive Sequence Libraries Preparation

The gDNA was digested by adding 0.005 U/μL DNase I; then, it was put at 15 °C for 2 h. The fragment size was checked by an agarose gel. Finally, they were marked with Dig-11-dUTP or Bio-11-dUTP dUTP.

The gDNA was renatured at 65 °C according to the renaturing time of T(s) = 1/M (mol/L), 20/M (mol/L), 60/M (mol/L), and 100/M (mol/L) (M is the final gDNA concentration (g/L) divided by 339 (g/mol)). Then, it was digested with S1 nuclease at 37 °C for 8 min. The enriched DNA sequences were added to a poly-A tail and cloned. The corresponding primers were designed by Primer 5.0, and the probes labeled with Dig-11-dUTP were used by PCR for hybridization.

### 4.3. Reverse Dot Blot Hybridization for Library Selection

All purified plasmids containing clone sequences were quantified in NanoVue PlusTM (GE Healthcare, Princeton, NJ, USA) and then diluted to a final concentration of 50 ng/μL. These plasmids were denatured by heating to 100 °C for 7 min and then quickly chilled in an ice/water for 10 min. The denatured plasmids were transferred onto the Amersham Hybond-NC nylon membrane (GE Healthcare, Life Sciences, Indianapolis, IN, USA). Then, 1 μL of each plasmid was spotted onto the membranes, and DNA was fixed to the membrane by UV crosslink using a StratalinkerTM UV Crosslinker (Stratagene, La Jolla, CA, USA). After fixation, the membrane underwent pre-hybridization for 30 min. The *T. arundinaceum* gDNA probe was labeled with digoxigenin-11-dUTP (DIG) using a DIG Nick Translation Kit (Roche Diagnostics). Hybridization was performed as described in the Instruction Manual of the DIG High Prime DNA Labeling and Detection Starter Kit I (Roche Diagnostics). Hybridization signals were subsequently detected and quantified by using the AxioVision measurement module of the Carl Zeiss Scope.A1 Imager fluorescent microscope (Carl Zeiss, Gottingen, Germany). Adobe Photoshop 6.0 was used to adjust the pictures.

### 4.4. Selection of the Barcode Probe from Cot-20 Library

In order to obtain a valid barcode sequence, we selected the sequences located at the centromeres, one end, and the two ends of the chromosome. From the sequencing results, we compared with the nucleotide database on the NCBI website to find the highest homology. Then, these sequences were compared with DNAMAN to obtain the conservative sequences as a candidate, and the corresponding 3 pairs of primer sequences were designed (Table 2). A 200–300 bp target band was obtained by amplifying the probe sequence using the genomic DNA of *T. arundinaceum* as the template.

Each 20 μL PCR reaction included DNA template, 1× LA Buffer, 10 μM of the primer, 20 μM dNTP and 0.5 U LA Taq. Meanwhile, the PCR conditions were 95 °C for 3 min, followed by 35 cycles of 95 °C for 20 s, 54 °C for 20 s, 72 °C for 30 s and final incubation at 72 °C for 6 min.

### 4.5. Fluorescence In Situ Hybridization

We selected the thick root tips according to the enzymatic method with some adjustment [42]. The section of root tips containing dividing cells was dissected and digested in an enzyme mixture (1% pectolyase Y23, 2% pectinase, 2% RS and 4% cellulase Onozuka R-10) for 4 h at 37 °C. After digestion, the root sections were washed in water and then washed in Carnoy’s fixative (ethanol: acetic acid = 3: 1) two times briefly. The root sections were carefully broken by using a pipette tip. The cell suspension was dropped onto glass slides, and another 10 μL acetic acid was dropped when the slide almost dried. The prepared slides were placed at room temperature overnight to make the chromosome aging. We put the slides in 2× SSC three times in 3 min. The slides were dehydrated in 75% and 100% ethanol each for 3 min; then, we added 70% FD at 70 °C to denature. The slides were dried, and we added the probes for hybridization.

## Figures and Tables

**Figure 1 ijms-23-06726-f001:**
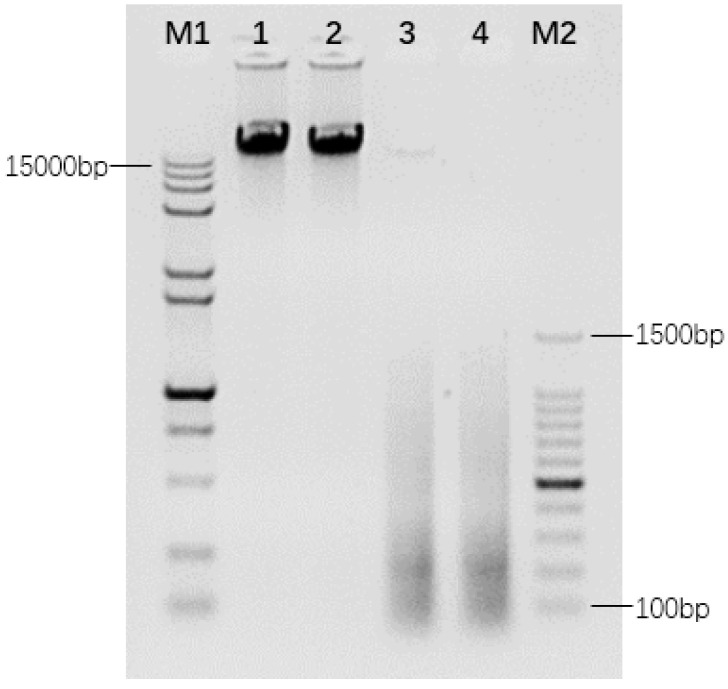
The electrophoresis result of *T. arundinaceum* genome DNA and interrupted genome DNA. M1: D15,000 + 2000 bp Marker; 1: Hainan 92–77 genome DNA; 2: Hainan 92–105 genome DNA; 3: The digested product of genome DNA from Hainan 92–77; 4: The digested product of genome DNA from Hainan 92–105; M2: 100 bp Marker.

**Figure 2 ijms-23-06726-f002:**
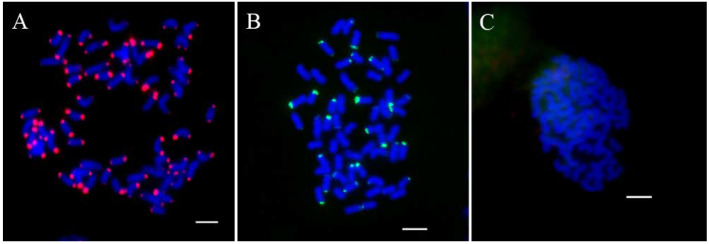
The signal expression to the different copied probes. (**A**): Two ends of chromosomes; (**B**): The end of a part of chromosomes; (**C**): No obvious signal. Scale bars = 5 μm.

**Figure 3 ijms-23-06726-f003:**
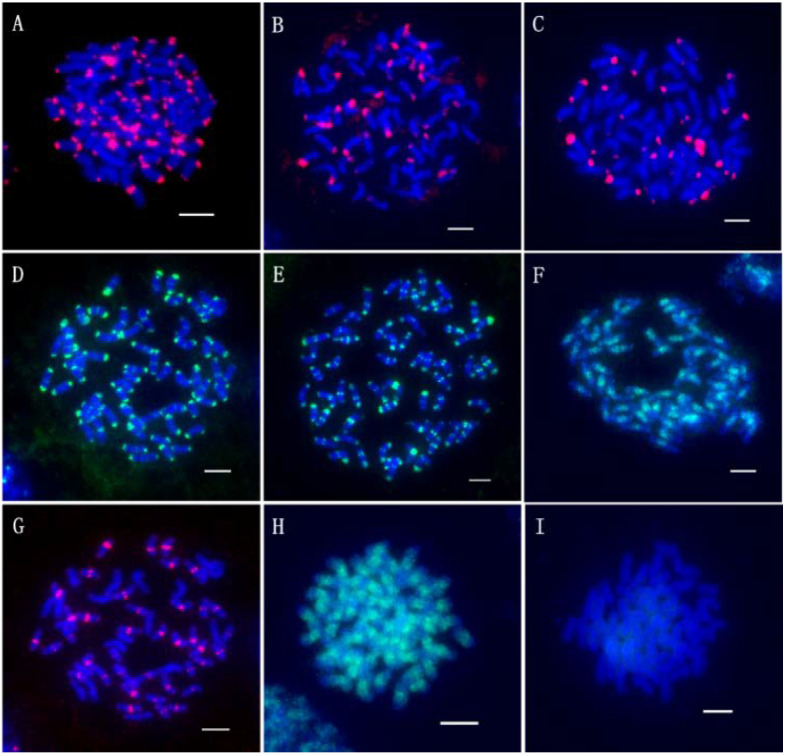
FISH result of the different probes. (**A**): Two ends of chromosomes; (**B**): The end of a part of chromosomes; (**C**): The end of most chromosomes and two ends of a part of chromosomes; (**D**): Two ends of chromosomes and the centromeric region of a part of chromosomes; (**E**): Two ends of chromosomes and the centromeric region of most chromosomes; (**F**): Diffuse distribute on all middle part of chromosomes; (**G**): The centromeric region of most chromosomes; (**H**): Diffuse distribute on all chromosomes; (**I**): No obvious signal. Scale bars = 5 μm.

**Figure 4 ijms-23-06726-f004:**
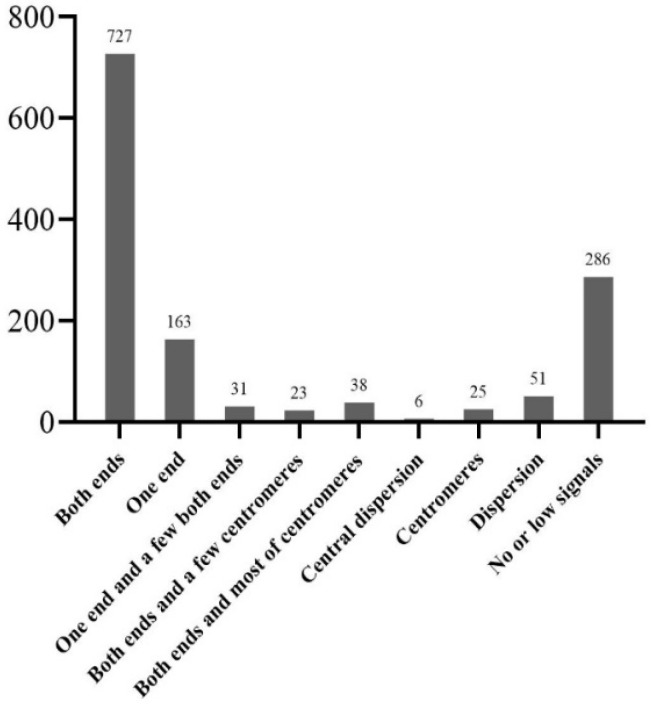
The statistic result of various types of the probe sequence.

**Figure 5 ijms-23-06726-f005:**
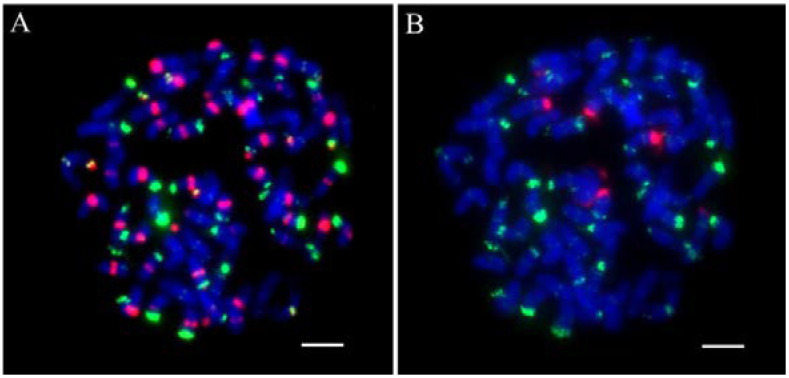
The result of repeat FISH. (**A**): The first round of FISH results; (**B**): The second round of FISH results. Scale bars = 5 μm.

**Figure 6 ijms-23-06726-f006:**
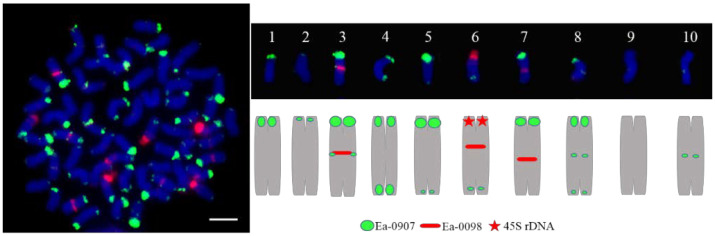
Chromosome karyotype of Hainan 92–105. Scale bars = 5 μm.

**Figure 7 ijms-23-06726-f007:**
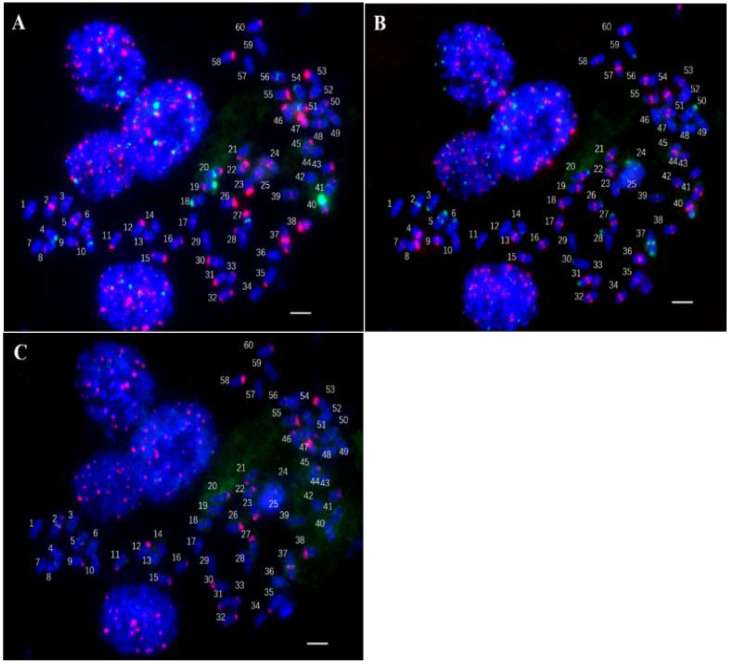
The result of three repeated FISH. (**A**): The first round of FISH results; (**B**): The second round of FISH results; (**C**): The third round of FISH results. Scale bars = 5 μm.

**Figure 8 ijms-23-06726-f008:**
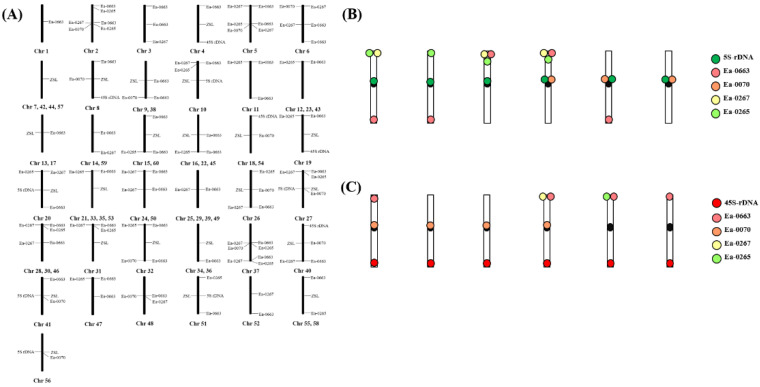
Distribution of the repetitive sequences in different chromosomes of *T. arundinaceum*. (**A**) The location of seven probes (Ea-0663, 5S rDNA, 45S rDNA, Ea-0267, Eacent, Ea-0070 and Ea-0265) on the chromosomes of *T. arundinaceum*; (**B**) The distribution of Ea-0663, Ea-0070, Ea-0267, and Ea-0265 on six homologous chromosomes that carried 5S rDNA loci; (**C**) The distribution of Ea-0663, EA-0070, Ea-0267, and Ea-0265 on six homologous chromosomes that carried 45S rDNA loci.

**Table 1 ijms-23-06726-t001:** The different types of chromosomes based on repetitive sequence probes.

Chromosome Number	1	2	3	4	5	6	7	8	9	10
Chromosome type	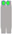	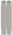	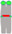	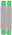	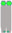	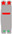	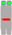	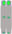	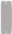	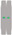
number	12	6	6	6	5	6	6	5	6	2

Note: Ea-0907: The end of the chromosome or centromeric region (green), Ea-0098: centromeric region (red), 45S rDNA: The end of the chromosome (red). Scale bars = 5 μm.

**Table 2 ijms-23-06726-t002:** The designed primer sequence by comparing.

Primer Name	Primer Sequence (5′—3′)
EaCent-F	CGGTTTGTTTGGAGACTTGC
EaCent-R	GCCCTAAATGATTTCTGAGCCTAT
EaST1-F	TTTTGGGACTCAGTTTCATTTC
EaST1-R	TGAAGACGCTAGAGTAGTATTTGTG
EaST2-F	TTACCATAAGCCACAAATC
EaST2-R	CATCTAAATACTCCACCCTAACT

## Data Availability

Not applicable.

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
