# Peer review of "Repetitive Sequence Barcode Probe for Karyotype Analysis in Tripidium arundinaceum"

_ijms, 2022, doi:10.3390/ijms23126726_

Round 1

Reviewer 1 Report

The authors apply the FISH methodology for the study of polyploidy in sugarcane. The phenomenon of polyploidy is very frequent in plants. This condition allows plants to have a greater amount of genetic material useful in genetic exchanges from generation to generation.

The method applied and discussed by the authors effectively describes this process. However, some points should be better discussed.

1) Why did the authors choose sugarcane? Why haven't they made a more complex study to validate the applied method in more plant species? Unfortunately, the FISH technique is not modern and many other technologies can replace it.

2) It is necessary for the authors to better discuss the usefulness of this technique. Indeed, why should we use the FISH technique for these analyzes?

3) The authors should also better discuss the importance of studying polyploidy in plants. Indeed this process is very frequent in the plants, and above all, it has been studied extensively with other approaches.

Author Response

Dear editor and reviewers:

We thank you for your time to process our manuscript “Repetitive sequence barcode probe for karyotype analysis in Tripidium arundinaceum”. All comments and suggestions from reviewers are very helpful for revising and improving our manuscript. In this revised version, changes to our manuscript within the document were all highlighted by using red colored text. Point-by-point responses to the reviewers are listed below this letter.

Reviewer 2 Report

The article "Repetitive sequence barcode probe for Karyotype Analysis in Tripidium arundinaceum" collects a great experimental work. The objective pursued by the authors to learn more about the origin and evolution and revealing the infiltration of genetic material in the process of polyploidization, was achieved.

The introduction, results have been correctly written. And the discussion reflects the results obtained

Despite being a great job, the presentation of this could be improved:

-in Figure 8, it is difficult to visualize the letters

- reference 41, I have not been able to obtain it in English. Therefore, the authors in the methodology should explain the procedure in detail.

- In addition, the conditions that have been used to amplify the DNA, with the different pairs of primers, should be reflected.

Author Response

Dear editor and reviewers:

We thank you for your time to process our manuscript “Repetitive sequence barcode probe for karyotype analysis in Tripidium arundinaceum”. All comments and suggestions from reviewers are very helpful for revising and improving our manuscript. In this revised version, changes to our manuscript within the document were all highlighted by using red colored text. Point-by-point Responses to the reviewers are listed below this letter.

Round 2

Reviewer 1 Report

The manuscript has improved considerably; the authors have finalized their paper following the indications suggested by the reviewers

I am very satisfied with the corrections and additions made by the authors.

In my opinion, the manuscript can now be published.